# Richelieu: Self-Evolving LLM-Based Agents for AI Diplomacy

Zhenyu Guan $^{\diamond}$, Xiangyu Kong$^{\clubsuit\dagger\boxtimes}$, Fangwei Zhong$^{\spadesuit\dagger\boxtimes}$, Yizhou Wang$^{\heartsuit\diamond}$

$^{\diamond}$ Institute for Artificial Intelligence, Peking University
$^{\clubsuit}$ College of Computer Science, Beijing Information Science and Technology University
$^{\spadesuit}$ School of Artificial Intelligence, Beijing Normal University
$^{\heartsuit}$ Center on Frontiers of Computing Studies, School of Computer Science,
Nat'l Eng. Research Center of Visual Technology, Peking University
$^{\dagger}$ State Key Laboratory of General Artificial Intelligence, BIGAI
$\boxtimes$Corresponding authors: `xykong@bistu.edu.cn`, `fangweizhong@bnu.edu.cn`

## Abstract

Diplomacy is one of the most sophisticated activities in human society, involving complex interactions among multiple parties that require skills in social reasoning, negotiation, and long-term strategic planning. Previous AI agents have demonstrated their ability to handle multi-step games and large action spaces in multi-agent tasks. However, diplomacy involves a staggering magnitude of decision spaces, especially considering the negotiation stage required. While recent agents based on large language models (LLMs) have shown potential in various applications, they still struggle with extended planning periods in complex multi-agent settings. Leveraging recent technologies for LLM-based agents, we aim to explore AI's potential to create a human-like agent capable of executing comprehensive multi-agent missions by integrating three fundamental capabilities: 1) strategic planning with memory and reflection; 2) goal-oriented negotiation with social reasoning; and 3) augmenting memory through self-play games for self-evolution without human in the loop. Project page: `https://sites.google.com/view/richelieu-diplomacy`.

## 1 Introduction

Diplomacy, a central element of international relations, is an intricate and multifaceted activity that lies at the heart of human society's most complex interactions. It requires various skills such as social reasoning, negotiation, and long-term planning to manage relationships and alliances among multiple parties. Mirroring this complexity, the Diplomacy game involves seven players to control European powers, presenting a challenging strategic landscape that demands advanced negotiation and strategic planning to succeed.

The AI community has shown an increasing interest in the deployment of AI agents to master such games [Shoker et al., 2023, Konya et al., 2023, Kramár et al., 2022, Duéñez-Guzmán et al., 2023, Mukobi et al., 2023, Kovač et al., 2023]. The recent breakthrough [Bakhtin et al., 2022] has turned into press diplomacy, which allows communication between players. However, the previous methods [Bakhtin et al., 2022] heavily rely on domain-specific human data, leading to its poor generalization to other scenarios/ applications. The question then arises: **Can we build an AI agent that excels in the art of diplomacy without relying on domain-specific human data?**

Recently, agents based on the Large Language Model(LLM) have emerged as a promising development for AI agents. The previous applications on personal assistants [Li et al., 2024b], robotics [Cheng

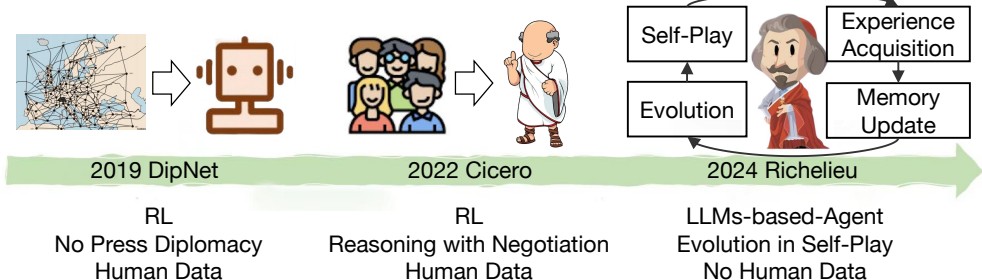

Figure 1: A new paradigm for building AI Diplomacy agent.

et al., 2024, Yang et al., 2023c], and video games [Wan et al., 2024] have shown the surprising ability of LLM-based agents in communication and planning, benefiting from the emergent ability of common sense reasoning, in-context/ few-shot learning, and sophisticated natural language processing on LLMs. However, diplomacy presents a unique set of challenges. It not only requires planning long-horizon strategic [Qi et al., 2024] and communicating with natural language, but also reasoning and adopting the complex social dynamics with partial observations, including gaining trust and reputation, building rapport, detecting deception, and assessing the reliability of other players.

In this work, we aim to make the first attempt to explore LLMs' potential to develop a human-like AI diplomacy agent. We name the agent Richelieu in memorizing a pivotal figure in European history who had enduring impacts on French politics, foreign affairs, and state building. To achieve this goal, we have identified four core and essential capabilities that are crucial for building an LLM-based societal agent.

1. **Social reasoning.** This is the basic function for a social agent to interact with others, particularly for adapting to the dynamic changes in the nation's intentions and relationships.

2. **Balance long- and short-term planning.** Diplomacy necessitates a careful balance between short-term tactics and long-term strategies. An effective AI agent must assess the immediate consequences of its actions alongside their potential long-term impacts.

3. **Memory management.** A robust memory system is a critical component of learning and improvement. The AI agent must be able to recall and integrate information from past negotiations and actions to inform its current and future decision-making processes. This endows the agent with the ability to evolve.

4. **Self-reflection.** An AI agent capable of profound reflection can analyze its own decisions, learn from its memory experience, and adapt its strategies accordingly.

By integrating these four capabilities, the agent can operate at the highest level of diplomatic sophistication, outperforming the state-of-the-art AI diplomats [Bakhtin et al., 2022].

Our contributions can be summarized in three-fold: 1) We introduced a new paradigm for building AI diplomacy agents, compared to previous work (Figure 1). The agent can self-evolve by generating experience via self-play games, without any task-specific human data. 2) We demonstrate the superior performance of our agent playing against the SOTA method, e.g., Cicero [Bakhtin et al., 2022], that relies on a large-scale human demonstration for training. 3) We further analyze the effectiveness of each module in our agent and the generalization of our agent in adopting different LLMs, such as GPT-4 and Llama 3.

## 2 Related work

**AI Diplomacy.** The diplomacy game involves seven players controlling different powers in Europe. In each turn, players can negotiate for cooperation before making moves to take as many supply centers as they can. Apparently, this challenging strategy task requires both complex negotiation skills and superior planning capability for player agents to achieve final victory. So far, most previous works on this task remain focused on the planning strategies (a.k.a. **No-Press Diplomacy** where no communication channels are allowed). The setting remains challenging considering its enormous

action space of $10^21$ to $10^64$ per turn (compared with Chess, which has much fewer than 100 actions per turn). No wonder existing efforts rely on human data to play the game. Among the methods, one typical research is DipNet [Paquette et al., 2019] which uses supervised and reinforcement learning. Based on DipNet, BRPI [Anthony et al., 2020], SearchBot [Gray et al., 2020], DORA [Bakhtin et al., 2021], and KL-Regularized search (Diplodocus) [Jacob et al., 2022] were conducted. Recently, research has also emerged for the full-setting of Diplomacy, i.e., **Press Diplomacy**, where players are allowed to negotiate with each other before making their moves in each turn. Such studies [De Jonge and Sierra, 2017][Bakhtin et al., 2022][Jaidka et al., 2024][Kramár et al., 2022] mainly benefit from the recent thriving language models. Specifically, notable advancements include policy iteration methods from DeepMind and Meta AI Research's equilibrium search agent [Jaidka et al., 2024]. However, Deepmind proposes to learn negotiation agents based on predefined contracts/protocols [Kramár et al., 2022]. And Meta AI's work, instead of one unified architecture, Cicero [Bakhtin et al., 2022] integrates a language model for negotiation and an RL model for planning respectively. Such separately trained models make it inconvenient for agents' continual evolution. Moreover, like no-press methods, these approaches heavily rely on human player data for agent training. Unlike these approaches, this paper delves into solving the negotiation and planning in one single self-evolving LLM-based agent model, without any pre-collected human expert training data.

**LLM-Based Agents.** With the emergence and growth of large language models (LLM), there is a growing trend in utilizing LLMs as fundamental controllers for autonomous agents[Wang et al., 2024c]. One wide application genre is LLM-based answering engines, which merely cover the negotiation aspects of Diplomacy. Such systems include HuggingGPT [Shen et al., 2023], GPT4Tools [Yang et al., 2023b] and ToT [Yao et al., 2023], etc. They leverage LLMs to manage AI models, use tools, implement policy iteration, and enhance problem-solving across various tasks. Related work including AutoGPT, AgentGPT, BabyAGl [Talebirad and Nadiri, 2023], Toolformer [Schick et al., 2023], and Visual ChatGPT aim to improve LLM's capabilities in task automation and tool usage. Reflexion, a framework that improves LLMs through linguistic feedback and episodic memory [Zhang et al., 2024a], facilitating better decision-making across diverse tasks is proposed. Besides [Wang et al., 2024d][Wang et al., 2023a][Wang et al., 2023b][Zhu et al., 2023][Yan et al., 2023] apply LLM agents to the complex planning tasks in the well-known open-world game Minecraft[Fan et al., 2022]. Unlike these LLM-based agents which only focus on the negotiation or planning ability respectively, the proposed approach involves a self-evolving scheme in a self-play game to handle both of them simultaneously.

## 3 Problem Statement

The Diplomacy game [Wikipedia, 2024, Calhamer, 1974] is set in pre-World War I Europe and involves each player (agent) representing one of the seven Great Powers of Europe, such as Germany, France, England, Italy, Austria-Hungary, Russia, and Turkey. Each player has a set of military units, including armies and fleets, which they can move and use to capture other supply centers. The ultimate goal for the agent is to control a majority of the total supply centers on the board by the end of the game's Fall phase. It's important to note that it is not won by eliminating other players or their units; it is won by controlling the requisite number of supply centers. This often involves forming and breaking alliances, negotiating, and sometimes betraying other players to achieve one's own goals.

In each turn, agent $i$ gets the current state $s_t \in S$, the actions of other players from the previous turn $\vec{a}_{t-1}^{-i}$, and the messages $\vec{m}_t^{-i,i}$ from other players during this turn's negotiations. The state $s_t^{,i}$ for the environment includes the ownership of each territory on the map by a particular country and where the armies of each country are located. Based on this information, the agent needs to engage in negotiations with other players, sending messages $\vec{m}_t^{i,-i}$ to chat with other players, and then take the actions $a_t^i$ in this turn. The possible actions an agent can take $a_t^i \in A$ are commands to the armies, such as moving into an adjacent territory, supporting another unit, or holding a position. Actions can also include diplomatic moves, such as proposing or withdrawing from an alliance, although these are less formalized in the game mechanics.[Paquette et al., 2019, Hill, 2014]

## 4 Self-Evolving LLM-based Diplomat

We have constructed a comprehensive framework with modules for memory management, social reasoning, strategic planning, negotiation, decision-making, memory update, and self-evolving to

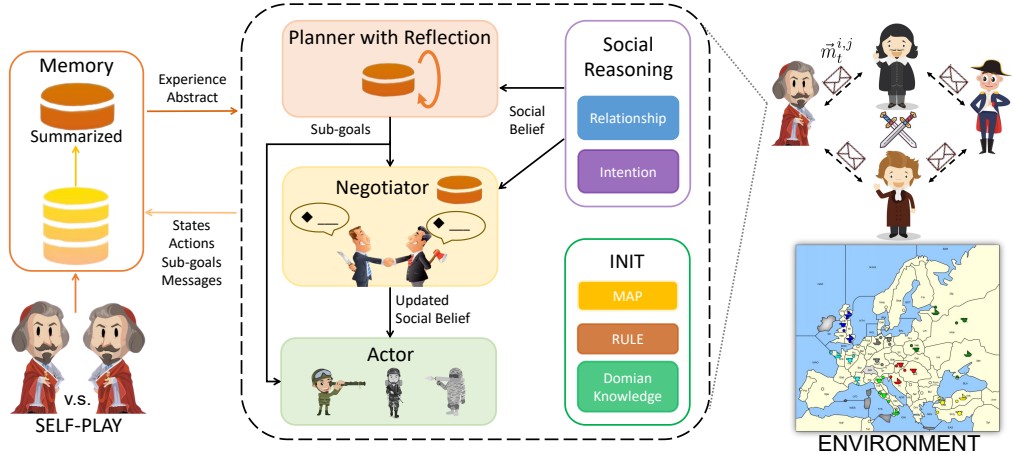

Figure 2: The framework of the proposed LLM-based-agent, Richelieu. It can explicitly reason social beliefs, propose sub-goals with reflection, negotiate with others, and take actions to master diplomacy. It augments memories by self-play games for self-evolving without any human annotation.

fully leverage the capabilities of LLMs. Richelieu starts by setting up with map details, game rules, domain knowledge, and the long-term goal.[Zhang et al., 2022, Wei et al., 2022, Wang et al., 2022a] At each turn, the agent will run in the following steps: 1) **Social Reasoning:** First of all, the agent undergoes a comprehensive analysis of the game state $s_t$ to build the social belief, including the intention of other players and their relationship $\vec{\phi_t} \in \Phi^n$.[Zhang et al., 2024c, Gürcan, 2024] 2) **Planner with Reflection:** Then, the agent proposes sub-goals $\chi_t^i \in X$ that is strategically aligned with the long-term goals $\Upsilon$, with the social belief and refining the proposed goal with experience $\vec{\eta_t} \in H^m$ abstract from the memory $M$ via self-reflection.[Wang et al., 2024b,e] 3) **Negotiator:** To achieve the sub-goals, the negotiator will start a dialogue session with some players, and evaluate their trueness $\vec{\psi_t}^{-i}$ by referring to their messages $\vec{m}_t^{-i,i}$, the current state $s_t$, their sincerity $\vec{\gamma_t}^{-i}$ and the experience $\vec{\xi_t}$.[Abdelnabi et al., 2023, Bianchi et al., 2024] 4) **Actor:** After negotiation, the actor decides its course of action $a_t^i$, based on the sub-goal $\chi_t^i$ and updated social state $s_{t+1}$, marking the end of that turn. 5) **Memory Management:** The state of the current turn $s_t$, the content of negotiations $\vec{m}_t$, the actions taken by all players $\vec{a_t} \in A^n$, and the sub-goals set forth $\chi_t^i$ are all logged within the memory as $\mu \in M$. This logged data serves as a historical experience, guiding Richelieu's subsequent actions in future turns [Hatalis et al., 2023, Zhang et al., 2024e]. 6) **Self-evolution:** The agent's evolution is highly dependent on the diversity of experiences stored in its memory. As this diversity grows, so does the agent's capability. Without human demonstrations, we employ multi-agent self-play games, i.e., our agents respectively control all the countries to simulate and acquire diverse experiences for self-evolving. Notably, the agent can further evolve during testing to adapt to different players.

## 4.1 Social Reasoning

There are no permanent enemies or allies. The relationship among countries is dynamically changing upon the evolving global state. However, it is difficult to determine the appropriate allies and enemies with partial observation. For example, there is uncertainty about the intentions of potential allies, which could lead to betrayal at pivotal moments. Consequently, we need to identify the intention and relationship of the current state by social reasoning to shape the social belief [Zhang et al., 2024c, Gürcan, 2024].

**1) Modeling Relationship:** Before setting sub-goals, Richelieu evaluates its relations with others, identifying enemies such as aggressive nations, vulnerable neighbors for expansion, and those with long-term potential threats. It also seeks out potential allies to counter these threats.[Sun et al., 2024, Zhang et al., 2024d] Simultaneously, Richelieu also tries to identify potential allies that could be instrumental in countering these adversaries. By isolating the analysis of inter-player relationships as a discrete element, Richelieu strategically exploits the actions of other players in subsequent stages of the game to reach its goals. **2) Inferring Intention:** The social belief is used by the planner, ensuring that its sub-goals are formulated with a comprehensive consideration of the behaviors and intentions

of other intelligent agents within the game. Richelieu's sub-goals will particularly emphasize those who are identified as potential adversaries or allies, fostering more effective collaboration with potential allies and participation in strategic opposition against adversaries. Furthermore, the insights gleaned from this analysis are instrumental in the subsequent negotiation phases. They are employed to assess the authenticity of the statements made by other players, as well as to aid Richelieu in reaching cooperative agreements [de Zarzà et al., 2023, He et al., 2024].

## 4.2 Strategic Planner with Reflection

The strategic planner specifies the sub-goals, which serve as an intermediary between immediate actions and the overarching goal of securing victory in the game. That is because we observe that LLMs are often characterized by their propensity to prioritize short-term gains in decision-making processes, with a notable deficiency in incorporating the future into their strategic calculations. [Renze and Guven, 2024, Zhang et al., 2024b]For example, it is common for a non-neighboring country to become too powerful. Formally, $\vec{\chi_t} \leftarrow SR(s_t, \vec{\phi_t}, \Upsilon)$ where $\vec{\chi_t} = (\chi_t^i, \chi_t^1, \ldots, \chi_t^n)$ represents the proposed sub-goals and other players' intention that we inferred, $\vec{\phi_t} \in \Phi^n$ represents the inferred relationship on the social belief. These goals may encompass a range of tactical considerations, such as the containment of a formidable rival's advancement or the strategic expansion in a particular direction to consolidate power.

**Reflection with Memory.** We further develop a reflection mechanism to enhance the rationality and effectiveness of our agent's sub-goals in achieving long-term goals.[Liu et al., 2024] This reflection mechanism relies on past experiences to critique and enhance proposed sub-goals. We employ a similarity-based function to find relevant historical experiences that match the current game state from its memory. This function considers two factors: goal similarity and state similarity, to select the most comparable experiences. The process can be written as: $\vec{\eta_t} \leftarrow h(s_t, \chi_t^i, M)$, where $\vec{\eta_t} \in H^m$. In practice, considering the limited context windows of LLM, we retrieve the most analogous experiences from the memory based on these metrics. Experiences with high evaluative scores reinforce successful strategies and support the continuity of existing sub-goals. On the other hand, lower scores indicate areas that need improvement and prompt the necessary adjustments. As our agent, Richelieu undergoes more training sessions, and hence its reflection ability gets improved. The growing pool of historical experiences consistently enhances its performance.

## 4.3 Negotiator and Actor

By chatting with other players, the goal of the negotiation is to update the social belief according to the received words and reach the sub-goal by manipulating others' intentions, such as securing cooperative agreements with other nations, terminating ongoing conflicts with a specific country, or deterring the formation of alliances directed against its interests.[Noh and Chang, 2024, Zhan et al., 2024] However, it is difficult to reach a consensus, as the interests and strategies of the various nations often conflict, and trust between players can be scarce, making it challenging to establish and maintain cooperative agreements. In this case, we argue that the negotiator should identify the true intentions and relationship of the opponent before generating words for the negotiation.

To fully utilize the power of LLMs, we construct a social reasoning flow for negotiation, as shown in Figure 3. During the negotiation process, we guide Richelieu to consider the veracity of what other players said and their true intentions, in conjunction with our established sub-goals and analysis of our relationships with other players, to negotiate and form alliances with potential allies and attempt to deceive enemies [Xia et al., 2024, Moghimifar et al., 2024].

To counteract the challenge of non-binding agreements and potential deception, we incorporate a discrete module dedicated to the assessment of the veracity of statements made by other players during negotiations. To determine the truthiness of other players' statements $\psi_t^j$, three main factors are considered. The most important is the consistency between the player's sub-goals $\chi_t^j$ that our agent inferred before and the intentions conveyed through his statements $m_t^{j,i}$. To aid the judgment, our agent also goes through the memory to retrieve the consistent experiences $\vec{\xi_t}$. Additionally, the player's overall honesty score $\gamma_i$ is taken into account. Hence, we get the truthiness of the opponent $j$: $\psi_t^j \leftarrow g(s_t, \chi_t^j, m_t^{j,i}, \vec{\phi_t}, \gamma_j, \vec{\xi_t})$, where $\vec{\xi_t} = w(s_t, m_t^{j,i}, M)$. With such a reasoning flow, our agent

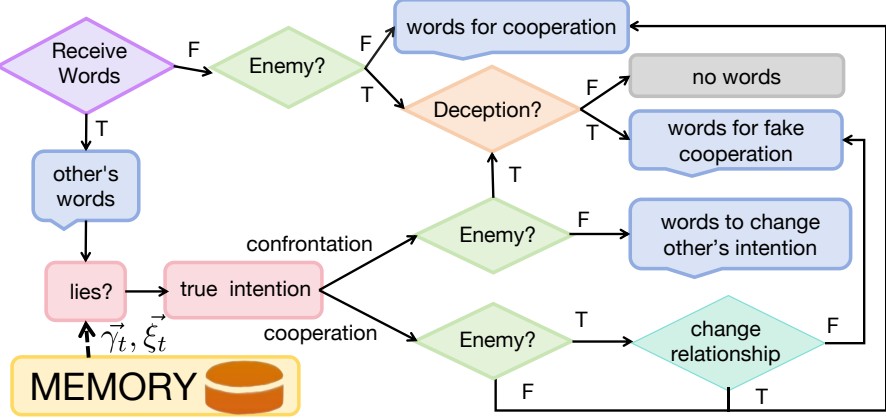

Figure 3: The social reasoning flow for negotiation. With the received words and memory, the agent will reason by answering the following questions: "Is the opponent lying?", "What is the true intention of the opponent?", "Is the opponent enemy?", "Is it necessary to deceive the opponent?", and "Is it necessary to change the relationship with the opponent?", and then generate the words accordingly for negotiation.

can adeptly navigate diplomatic discourse. After the negotiation, the actor will get the updated social beliefs and choose a specific action for the army.

## 4.4 Memory Management and Evolution in Self-Play Games

Memory is the foundation of the framework that accumulates the historical experience of the agent and summarizes them for other modules [Gao and Zhang, 2024, Li et al., 2024a, Yu et al., 2024, Hou et al., 2024]. It supports other modules, such as planner and negotiator, to provide long-tail experiences.

**Experience Management.** Specifically, the memory module is tasked with the acquisition and archival of historical data, encompassing the observed game state $s_t$ at each turn, its sub-goals $\chi_t^i$, the messages during the negotiation $\vec{m}_t$, and the actions of all the players $\vec{a}_t$. Subsequently, the raw experience is summarized in a shorter content with an evaluation $\lambda_t \in \Lambda$ of the proposed sub-goals and an assessment of the credibility of other players $\gamma_j \in \Gamma$. $\lambda_t$ serves to reflect upon the agent's sub-goals. It evaluates whether sub-goals are reasonable based on the subsequent state and long-term goals $\Upsilon$. As the game progresses, it is continuously updated in response to changes in the state $\lambda_t \leftarrow f(\chi_t^i, \Upsilon, \vec{s})$, where $\vec{s} = (s_t, s_{t+1}, \dots s_T)$. The formula represents the update of the evaluation $\lambda_t$ for the sub-goal in turn $t$ by the memory in turn $T$. The updates will cease when there is a fundamental change in the sub-goal compared to the goal at turn $t$. This prevents subsequent decisions from impacting the assessment of the current decision-making. We employ $\gamma_j \in \Gamma$ to evaluate the credibility of player $j$ and utilize $\tau_t^j \in \{0, 1\}$ to denote the truthfulness, i.e., whether the statements made by the player $j$ during the negotiation process at time $t$ are truthful. The truthiness of player $j$'s statements is updated according to the memory from the previous turns, $\tau_t^j \leftarrow T(s_t, s_{t+1}, a_t^j, m_t^{j,i})$. The credibility of player $j$, $\gamma_j$ will be updated based on player $j$'s statements $\tau_t^j$, written as $\gamma_j \leftarrow p(\gamma_j, \tau_{t-1}^j)$. Players' credibility $\vec{\gamma}$ is a short-term memory that is applicable only to the current turn. Other data collected or generated constitutes long-term memory. These data will be combined to form a history $\mu \in M$, and then is incorporated into memory.

**Acquisition Experience via Self-Play Games.** self-play training mechanism has been widely applied on training agents via reinforcement learning [Zhong et al., 2019, 2021, Wu et al., 2022]. However, it is hardly explored in LLM-based agents. For LLM-based agents, a self-play game allows the agent to accumulate more experiences for self-evolution [Liu et al., 2024, Zhang et al., 2024a]. As self-play continues, the acquisition of new and better historical experiences by the agent will diminish. This means that the agent's capabilities will not improve indefinitely. At the same time, as the memory grows, selecting appropriate historical experiences becomes a new challenge. The chosen $m$ experiences $\vec{\eta}_t$ may be almost identical, which could actually reduce the amount of useful information available to Richelieu. After self-play training, when Richelieu is faced with a certain

Table 1: The results of our method playing against Cicero.

| Model | Win↑ | Most SC↑ | Survived↑ | Defeated↓ | Model | Win↑ | Most SC↑ | Survived↑ | Defeated↓ |
|-------|------|----------|-----------|-----------|-------|------|----------|-----------|-----------|
| Richelieu_1 | 6.20% | 9.40% | 38.90% | 45.50% | Richelieu_1 | 6.30% | 7.90% | 39.40% | 46.40% |
| Richelieu_2 | 6.60% | 7.80% | 40.80% | 44.80% | Richelieu_2 | 6.60% | 8.30% | 41.20% | 43.90% |
| Richelieu_3 | 7.10% | 9.30% | 39.90% | 43.70% | Richelieu_3 | 7.20% | 8.70% | 41.70% | 42.40% |
| Richelieu_4 | 7.40% | 8.00% | 40.20% | 44.40% | Cicero_1 | 5.80% | 6.70% | 41.20% | 46.30% |
| Cicero_1 | 5.90% | 6.50% | 41.50% | 46.10% | Cicero_2 | 6.50% | 7.20% | 42.50% | 43.80% |
| Cicero_2 | 6.30% | 7.20% | 42.50% | 44.00% | Cicero_3 | 6.00% | 7.00% | 41.60% | 45.40% |
| Cicero_3 | 5.90% | 7.00% | 41.60% | 45.50% | Cicero_4 | 6.10% | 7.20% | 42.30% | 44.40% |
| **Richelieu** | **6.83%** | **8.63%** | **39.95%** | **44.60%** | **Richelieu** | **6.70%** | **8.30%** | **40.77%** | **44.23%** |
| **Cicero** | **6.03%** | **6.90%** | **41.87%** | **45.20%** | **Cicero** | **6.10%** | **7.03%** | **41.90%** | **44.98%** |

state, it can draw on a larger pool of similar historical experiences. Diverse experiences enable the agent to reflect more comprehensively on the strategies it currently devises, leading to a stronger optimization of decision-making. As shown in Figure 5, Richelieu's performance against Cicero [Bakhtin et al., 2022] becomes better with increasing training iterations. With the accumulation of experiences, Richelieu's win rate exhibits a steady increase with accumulated training iterations, ultimately plateauing at a stable performance level. In contrast, the defeated rate shows a consistent decrease, approaching an asymptotic value. These observations confirm the effectiveness of self-play in Richelieu's evolution.

# 5 Experiment

In the experiments, our goal is to answer the following questions: 1) **Mastery of Non-Press Diplomacy**: Can our agent master the non-press diplomacy against baselines? 2) **Competing with State-of-the-Art**: Can our agent surpass the performance of the current state-of-the-art agents in press diplomacy? 3) **Compatibility with LLMs**: Can our self-evolving framework be compatible with different LLMs? 4) **Contribution of Each Module**: Do the individual modules within our framework contribute to the overall improvement of our agent's performance?

## 5.1 Experimental Setup

**Environment.** The widely-used open source Diplomacy game platform introduced by [Paquette et al., 2019] is adopted for evaluating Richelieu against other models. It is easy to switch between no-press (with negotiation between players) and press (no negotiation between players) games based on this platform, facilitating a comparison of both settings. The platform also contains over 10,000 human game data on which previous approaches are used. Note that our method does not need them. In each episode, a model will host one randomly selected country to compete against countries controlled by other methods. It wins if it occupies all the supply centers and loses and vice versa.

**Evaluation Metrics.** We evaluate the models based on the results of multiple rounds of games. In each round, the model is randomly assigned a country to control. Typically, 1000 rounds are played to obtain the average results. We evaluate the models in two metrics. One is based on the win rate, Most SC rate, survived rate, and defeated rate. There are four possible outcomes for each country in the game. If a country loses all its supply centers (SC), it is eliminated and recorded as "defeated". If a country occupies 18 or more out of 34 supply centers, the game ends, and that country is recorded as "win", while other countries are recorded as "defeated". In other cases, the game ends in a draw. The country with the most supply centers is recorded as "Most SC", the countries that have been eliminated are recorded as "defeated", and the other countries are recorded as "Survived". The other is based on the scores obtained by the models after multiple rounds of competition. To compare the capabilities of multiple models, we use C-Diplo Argir[Archer, 2024], a scoring system. This system is used in many international diplomacy competitions. The scoring method is as follows: If a player wins by occupying 18 or more supply centers, the player scores 93 points, and each of the other six players scores 1 point. If the game ends in a draw, the player with the most centers scores 37 points. The second player with the most centers scores 14 points. The third player with the most centers scores 7 points. Each player scores 1 point per center owned. Each player also scores 1 point for participating. In this way, regardless of the game outcome, a total of 99 points will be distributed among the players in each game.

**Baselines.** We select six previous models as baselines for comparison. Among them, Cicero[Bakhtin et al., 2022] by Meta is a diplomacy model with a negotiation module. The SL-DipNet and RL-DipNet [Paquette et al., 2019], the BRPI [Anthony et al., 2020], the SearchBot [Gray et al., 2020], and the DORA[Bakhtin et al., 2021] are no-press diplomacy models. We also build an LLM-based agent, AutoGPT [Yang et al., 2023a]. In experiments, we set a temperature of 0.3 to ensure a relatively stable generation of LLM policies. The overall reasoning framework also ensures the stability and consistency of the AI agent's performance.

## 5.2 Results

**Massively Play with Baselines on no-press setting.** We let Richelieu compete with the other six models including Cicero[Bakhtin et al., 2022], SL-DipNet and RL-DipNet [Paquette et al., 2019], BRPI [Anthony et al., 2020], SearchBot [Gray et al., 2020], and DORA[Bakhtin et al., 2021] on No-Press Diplomacy, in which players make moves without communication. Figure 4 indicates that Richelieu outperforms other previous models relying on human game data. In contrast, Richelieu does not need such data but outperforms these methods by a clear margin, which demonstrates the outstanding planning capability of Richelieu.

**Play against Cicero on press setting.** We also evaluate Richelieu through competition against Cicero in the challenging scenario where negotiation is enabled. Specifically, we randomly assign three countries to one model and the remaining four to another. After playing several

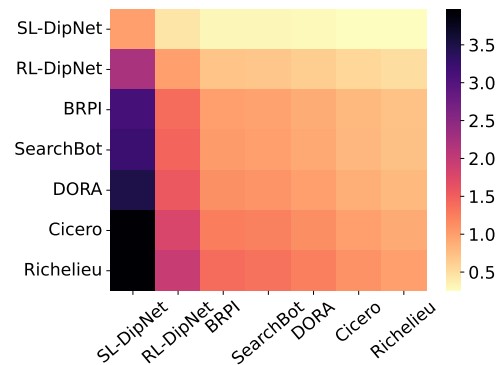

Figure 4: The relative scores among 7 different agents when massively playing on the no-press setting. Each point shows the ratio of the model's score on the vertical axis to the score gained by the model on the horizontal axis.

rounds of the game, the win rate, most SC rate, survived rate, and the defeated rate is calculated using a weighted average for evaluation. Table 1 demonstrates the competitive performance of Richelieu in comparison to Cicero. Richelieu's win rate is approximately 0.7% higher than Cicero's. If the Most SC rate is also taken into account, Richelieu is about 2% higher than Cicero. At the same time, Richelieu's loss rate is also 0.6% lower. According to our scoring system, Richelieu's score is about 10% higher than Cicero's. This is nontrivial especially when Richelieu is trained in a self-play game without humans and the opponents are trained with the data from human players.

Although Richelieu's win rate improvement compared to Cicero is not significant, the relative value of the improvement is quite large. Moreover, the main reason for the modest improvement is that in the seven countries, there are three or four controlled by Richelieu with similar abilities, which often results in the game ending in a draw. Moreover, we observed a large gap by comparing the scores the agents gained in the massively play with baselines on the no-press setting shown in Figure 4. Our agent's score is about 10% higher than Cicero's.

**Play against AutoGPT on press setting.** We further built an LLM-based agent using AutoGPT and compared it with our agent. In the testing, we randomly select three countries to be controlled by Richelieu, and the other four countries to be controlled by AutoGPT. Note that the agent controls each country independently. The results are shown in Table 2. We can see that our model significantly outperforms the off-the-shelf reasoning framework for LLM-based agents.

Table 2: The results of Richelieu playing against AutoGPT.

| Model | Win↑ | Most SC↑ | Survived↑ | Defeated↓ |
|---|---|---|---|---|
| Richelieu_1 | 9.30% | 18.20% | 37.90% | 34.60% |
| Richelieu_2 | 9.90% | 19.40% | 37.70% | 33.00% |
| Richelieu_3 | 8.10% | 17.40% | 39.20% | 35.30% |
| AutoGPT_1 | 1.20% | 4.60% | 32.40% | 61.80% |
| AutoGPT_2 | 1.20% | 4.20% | 34.40% | 60.20% |
| AutoGPT_3 | 1.50% | 4.00% | 32.50% | 62.00% |
| AutoGPT_4 | 2.60% | 3.60% | 32.30% | 61.50% |
| **Richelieu** | **9.10%** | **18.33%** | **38.27%** | **34.30%** |
| **AutoGPT** | **1.63%** | **4.10%** | **32.90%** | **61.37%** |

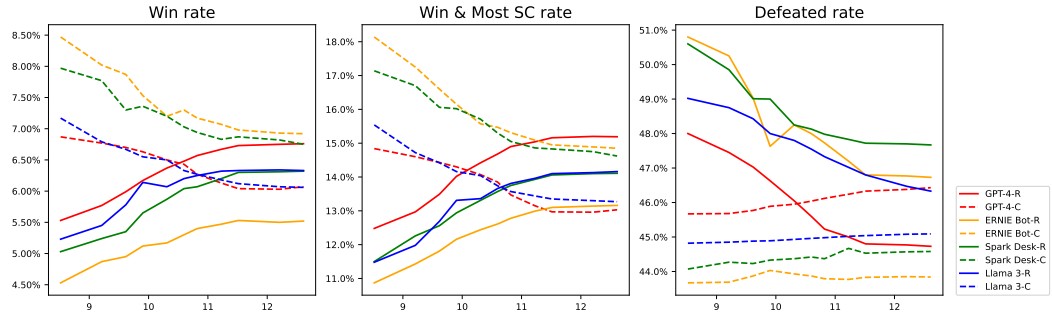

Figure 5: Richelieu modules benefit different LLMs. The solid line represents the experimental results for Richelieu, while the dashed line corresponds to Cicero. Different colors are used for different LLMs. The horizontal axis represents the logarithm of the number of training sessions, and the vertical axis denotes the rate.

Table 3: Ablation study: average results of 3 Richelieu vs. 4 Cicero.

| Modeling others | sub-goals | Negotiation pipeline | Reflection with Memory | Self-play | Win ↑ | Most SC↑ | Survived↑ | Defeated↓ |
|---|---|---|---|---|---|---|---|---|
| | | | | | 0.4% | 0.7% | 4.3% | 94.6% |
| ✓ | | | | | 0.7% | 1.2% | 10.6% | 87.5% |
| ✓ | ✓ | | | | 3.3% | 4.7% | 26.7% | 65.3% |
| ✓ | ✓ | ✓ | | | 3.8% | 5.8% | 33.1% | 57.3% |
| ✓ | ✓ | ✓ | ✓ | | 5.2% | 6.6% | 39.5% | 48.7% |
| ✓ | ✓ | ✓ | ✓ | ✓ | **6.7%** | **8.5%** | **40.4%** | **44.4%** |

**Generalization of self-evolving framework to different LLMs.** To demonstrate the effectiveness of our framework in a variety of LLM, we conducted experiments using four models: GPT-4, ERNIE Bot, Spark Desk, and Llama 3. As the number of training iterations increases, Richelieu's win rate steadily improves while the defeated rate declines, ultimately reaching a relatively stable outcome. This suggests that our self-play method is effective. After training, the win rate using GPT-4 increased from 1.5% lower than Cicero's to about 0.7% higher than Cicero's. The win rate using Llama 3 increased from 2.3% lower than Cicero's to almost equal to Cicero's. The win rates using Models Spark Desk and ERNIE Bot increased from 3% and 4% lower than Cicero's to 0.7% and 1.6% lower than Cicero's, respectively. The experimental results show that, despite variations in Richelieu's performance due to the inherent differences in the capabilities of these LLMs, as illustrated in Figure 5, our framework and training approach significantly enhance the capabilities of all LLMs.This indicates the generalization of a self-evolving framework to various LLMs. To demonstrate the effect of the memory from the self-play game on our agent, we found two turns with similar states in different rounds, one before self-play and the other after. The cases are shown in Appendix B.1.

**Ablation Study.** We conduct comprehensive ablation studies on Richelieu by analyzing the benefit of incorporating Richelieu's various modules, like planners or memory, into basic LLMs. The results are shown in Table 3. As illustrated in Figure 5, while the enhanced alignment in LLMs indeed boosts performance (GPT-4 is better than others), we observed that a vanilla GPT-4 still falls short in AI diplomacy without our framework, as can be seen in Table 3. Richelieu's performance has markedly improved with the integration of each module, demonstrating its ability to leverage other players' actions in decision-making while balancing short-term and long-term benefits. Its negotiation skills have also enhanced significantly, enabling it to clearly communicate intentions to cooperate and avoid deception. Moreover, the self-play experience further boosts Richelieu's performance. These findings suggest that while alignment in LLMs is essential, our approach is crucial for unlocking models' potential in social simulation.

# 6    Conclusion

In this paper, we introduce Richelieu, a self-evolving LLM-based agent for AI diplomacy. Our model enables hierarchical planning for multi-agent tasks and utilizes a memory module for reflective optimization. Our model does not require human data and can evolve through self-play. It ultimately outperforms existing models like Cicero in the Diplomacy. Our ablation study demonstrates the effectiveness of the modules we have established. By conducting experiments using different LLMs, we validate the generalization of our framework to various LLMs. We believe that the use of LLM-based agents will become an effective approach in social science in the future.

# 7    Limitations and Future Work

Our study is subject to certain limitations. We utilize diplomacy as the platform for constructing our model. However, the space of actions within diplomacy is constrained, whereas the decision-making space in real-world diplomacy is virtually boundless. In diplomacy, apart from the information exchanged between players during the negotiation, all other information is public and certain. Conversely, real-world diplomacy operates within a framework of incomplete information.

Our framework is capable of applying to most social interaction tasks. Most components in our framework can be easily generalized to a new task by modifying the content. Social reasoning enables the agent to handle complex and dynamic social relationships. The negotiation pipeline opens the potential of communicating with others to prob the other's mind or reach a consensus. The hierarchical strategy with reflection enhances the ability to handle long-term planning. The self-evolving mechanism (reflection with self-play memory) further improves the overall performance without manual supervision. These modules cover most of the challenges in multi-agent interactions. The potential applications of such an AI agent are vast, ranging from simulated diplomatic environments to real-world assistance and analysis. In future research, we intend to develop a more realistic game space, characterized by incomplete information and multi-player games, to enhance and refine our model further. We will also extend the framework to other multi-agent scenarios, including embodied interactions [Zhong et al., 2023, Ci et al., 2023, Chen et al., 2023], sensor networks [Wang et al., 2022b, Xu et al., 2020, Pan et al., 2022, Li et al., 2020], and video games [Wang et al., 2024a, Ma et al., 2024]. This framework can also be employed to develop various applications, such as recommendation [Huang et al., 2023, Chen et al., 2024, Huang et al., 2024, Hong et al., 2024], business negotiation [Hua et al., 2024], and education [Shea et al., 2024].

# 8    Ethical Consideration

The method proposed in this work has the potential for positive uses like enabling AI agents to emerge in cooperation via negotiation or avoiding being fooled by fake promises (or helping humans do so). However, negative cases can also arise if the technique is used for possible fraud activities. Fortunately, there is research [Bakhtin et al., 2019][Zellers et al., 2019] dealing with such scenarios. And we also urge for more research efforts in this field to foster safe applications of similar technologies.

## Acknowledgements

This work was supported by the National Science and Technology Major Project (2022ZD0114904), NSFC-6247070125, NSFC-62406034, NSFC-62406010, the State Key Lab of General Artificial Intelligence at Peking University, Qualcomm University Research Grant, and Wuhan East Lake High-Tech Development Zone, National Comprehensive Experimental Base for Governance of Intelligent Society.

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

# A Implementation Details

## A.1 Rules of Diplomacy Game

- You need to occupy as many supply centers as possible. If you occupy 18 or more supply centers, you will win the game directly. If you lose all your supply centers, you will be eliminated immediately.

- The units consist of armies and fleets. Armies can only move to adjacent areas, while fleets can move to adjacent sea zones or coastal areas and can move along the coast.

- To occupy a supply center, your units must move into that area in the autumn.

- When a unit moves to an area, if another unit is in the destination or if other units are also moving to that destination, the move fails, resulting in a standoff. In such cases, you can seek support from units in adjacent areas to the destination. If another unit moves into the region from which support is coming, the support is cut off. The unit with the most support moves into the area, while other units must retreat to an adjacent province or disband. If there is no place to retreat, the unit must disband. Fleets can transport armies across sea zones from one coastal region to another. However, if another fleet moves into that sea zone, the transport is cut off.

- The number of units a country can have cannot exceed the number of supply centers it controls. If the number of supply centers decreases, excess units must be disbanded. Each autumn, new units can be built at supply centers. Coastal supply centers can produce fleets or armies, while others can only produce armies. [Hill, 2014]

## A.2 Domain Knowledge

Richelieu can adopt a strategy of allying with distant countries while attacking neighboring ones to occupy adjacent territories and achieve rapid expansion. Richelieu should pay attention to the Balance of Power by forming alliances with other countries or supporting weaker states to prevent any single country or alliance from becoming too powerful. [David, 2014] To this end, Richelieu can also adopt a strategy of attacking distant countries while allying with nearby ones, sacrificing short-term benefits to avoid the emergence of future hegemonic states that could threaten his own survival. When facing multiple enemies, Richelieu can find ways to divide other countries and incite wars among them. Whether in offense or defense, Richelieu should actively choose suitable allies. Richelieu can also introduce a third party to achieve goals such as ceasefire, alliance, or joint attack. To achieve alliances or ceasefires, Richelieu can sacrifice some interests to the other party as long as the ultimate benefits are greater. Others may lie and deceive [Kostick, 2015]; their words in negotiations are not binding. Richelieu must avoid being deceived or betrayed. At the same time, Richelieu can also actively deceive others to achieve his own goals.[Richard, 1979, Allan, 1975]

## A.3 Social Reasoning

We conduct an experiment to evaluate the success rate of the agent $i$ successfully identifying the social relationships and inferring others' intentions. As the baselines do not explicitly model the relationship and intention, we can not directly access the ground truth for evaluation. Instead, we let all players use our agent but with different LLMs, i.e., 4 countries use GPT-4 and 3 countries use Llama 3. The accuracy is reported in Table 4. We can see that the accuracy of social reasoning is consistent with the overall performance of the agent, indicating the effectiveness of social reasoning.

Table 4: The success rate to identify the social relationship and infer others' intentions.

|  | GPT-4 | Llama 3 |
| --- | --- | --- |
| relationship | 85.74% | 85.52% |
| intention(sub-goal) | 74.67% | 74.11% |

## A.4 Prompt Templates

For the convenience of reproducing the results of the experiments of this paper, here we give the prompt template of different modules of Richelieu.

**1**) INIT

```
1   You will control {country} and compete with six other countries
     on the map for supply centers.
2   The map consists of different regions and sea areas. Their
    adjacency relationships are shown in the matrix. The numbers
    for the regions and sea areas are ......
3   Different regions are occupied by different countries. The
    ownership of the regions is shown in the matrix.
4   The region Berlin, ........ are supply centers.
5   You need to follow these rules ......
6   To help you achieve victory, these diplomatic strategies might
    be of assistance. ......
```

**2**) Social Reasoning

```
1   France occupies Portugal Ruhr, Paris, Burgundy, ......
2   France has armies in Brest, Belgium, ...... And France has
    fleets in Mid Atlantic, England Channel, ......
3   England ......
4   ......
5   Based on the current state, what do you think are the current
    strategic intentions of the other countries?
6   Which country do you think needs to be attacked or weakened the
     most right now?
7   And which country do you think is most suitable for you to ally
     with in order to deal with this country?
```

**3**) Planner with Reflection

```
1   In the current state, with {ally and enemy}, what sub-goal do
    you think should be set for {country} ?
2   I have found some useful historical experiences for you. Please
     reflect on and optimize your sub-goal based on these
    historical experiences.
3   The sub-goal you formulated when {state} was to {sub-goal}. The
     eventual result was {future}. The evaluation for this sub-goal
     is {score}.
```

## A.5   Project Website

https://sites.google.com/view/richelieu-diplomacy

# B   Cases

## B.1   Cases of the Effect of the Memory from Self-Playing and Collaboration

As is shown in Figure 6, Richelieu controls France. In the two cases, France is at war with Austria. However, Russia is on the verge of victory in its war against Turkey, which will lead to significant territorial expansion for Russia. France and Russia currently do not share a border, are not at war, and have no conflicts of interest.

In case 1, before the self-play, in the current turn, Richelieu failed to realize the potential threat from Russia and continued to attack Austria. Thus, in this round, Russia ultimately won the game. Figure

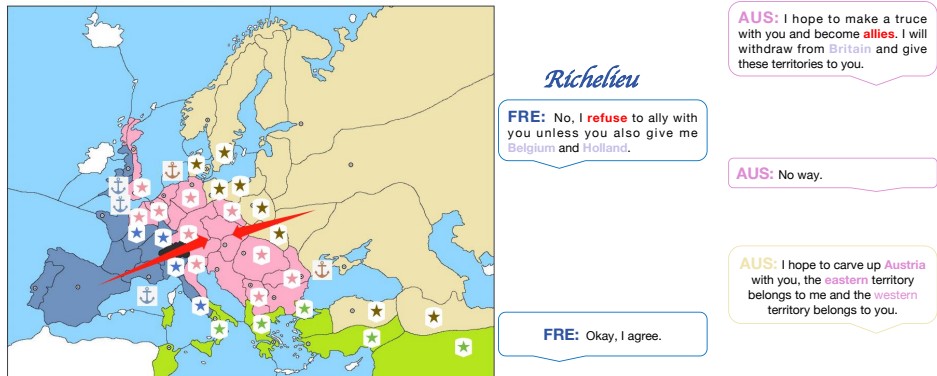

(a) Case1: The agent **without** self-play memory tends to ignore long-term gains.

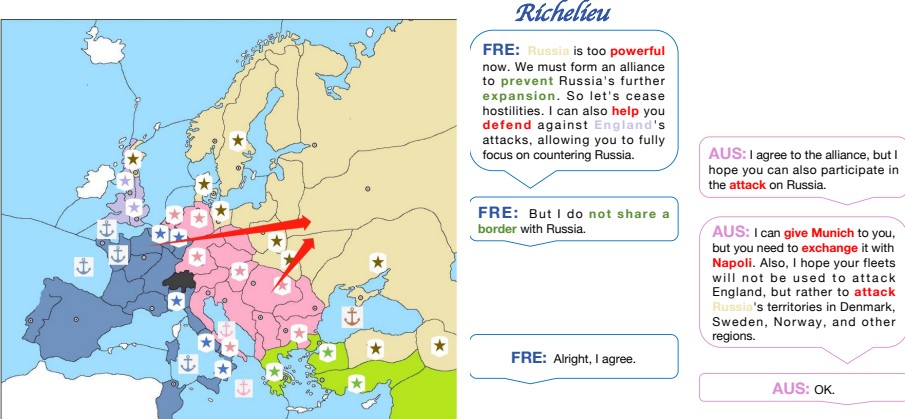

(b) Case2: The agent **with** self-play memory tends to consider long-term gains.

Figure 6: Case of self-playing before and after comparison.

6(a) shows the state and the negotiation before the self-play, where we rejected Austria's request for an armistice and alliance.

After self-play, using the historical experience from the memory module, Richelieu adjusted his strategy. Richelieu foresees Russia becoming the most threatening enemy in the future and sets a sub-goal of weakening Russia, allying with Austria and Turkey, and attacking Britain. Figure 6(b) shows the state and the negotiations after self-play, where we actively sought an armistice alliance with Austria to make Austria concentrate their forces against the Russian attack. In the subsequent negotiation phase, Richelieu proactively proposes ending the war with Austria, despite holding an advantage in this conflict. Richelieu promises Austria that if it ceases hostilities and attacks Russia, Richelieu will assist Austria in defending against any attacks from England. The negotiations are successful. Austria accepted Richelieu's proposal, and the two countries reached an agreement to exchange the supply centers of Napoli and Munich. During the action phase, Austria moves its troops from Venice to Apulia in preparation for capturing Napoli in the next turn, while the rest of its forces are repositioned to the eastern regions bordering Russia to defend against Russian attacks and compete for supply centers. French units occupy Munich and prepare to advance on Russian territories such as Berlin. Meanwhile, French units support Austria in the Holland and Belgium regions. In this round, we ultimately achieved a better result——Most SC. This is also a great example that highlights our model's ability to collaborate effectively with other players.

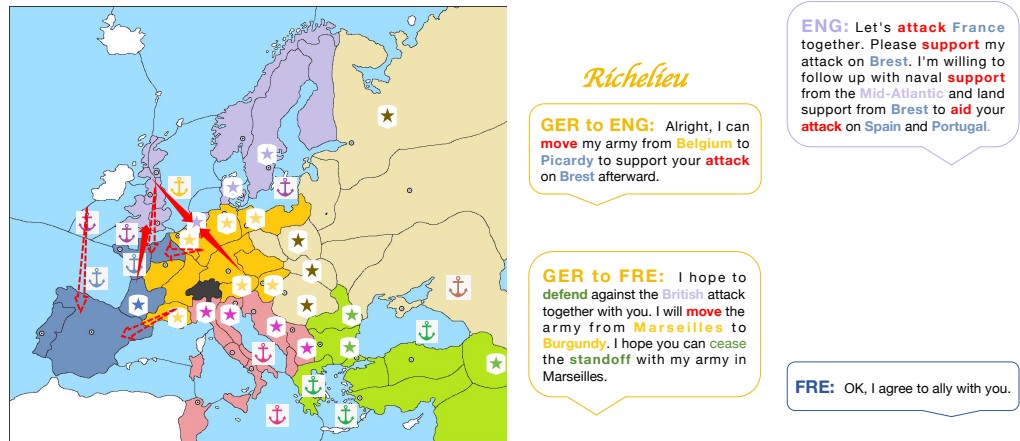

Figure 7: An example case of avoiding being deceived by other countries during negotiations.

## B.2 Case of Avoiding Deception

As shown in Figure 7, Richelieu controls Germany. During the negotiation phase, England proposed a ceasefire to Germany and invited Germany to ally to attack France jointly. England hoped to cease the war with Germany in Holland and Belgium. Subsequently, German units supported England in attacking Brest, and then England utilized its fleets to assist Germany in attacking Spain and Portugal. Richelieu suspected that England was deceiving Germany, as England was likely to attack territories in the north such as Belgium and Berlin after German units were redirected to support Brest. Therefore, we pretended to accept England's alliance proposal during the negotiation process. However, at the same time, we sought out France and expressed our willingness to cease hostilities, allowing France to focus entirely on defending against England's attacks. In the action phase, England's actions confirmed Richelieu's suspicions. England attacked Belgium from Holland, but because Richelieu didn't move units in Belgium, England's attack failed.

## C More application

Our modules cover most of the challenges in multi-agent interactions, e.g., economic games, and daily interactions. To prove that our framework is capable of applying to most social interaction tasks, we further adopt our framework to a werewolf game. The results demonstrate our reasoning framework achieves comparable results to the other methods. To be specific, in the experiment, we let our agent play as a werewolf in a seven-player game, where there are two werewolves, one witch, one seer, one guard, and two villagers. The experimental results show that the win rate of our agent is 59.2%, even without applying the self-play game in the current version. For comparison, the strongest specifically designed LLM-based agent achieved 65% win rate [Xu et al., 2023]. This proves that our model can be applied in more scenarios and achieve results comparable to those of specially designed models.

