# OpenReview forum: "Richelieu: Self-Evolving LLM-Based Agents for AI Diplomacy"
_NeurIPS.cc/2024/Conference — NeurIPS 2024 poster_

### Official Review · Reviewer_36qq · 2024-06-30

**Soundness:** 3
**Presentation:** 3
**Contribution:** 2
**Rating:** 7
**Confidence:** 4

**Summary:**

The paper introduces an LLM agent that can solve the game of Press Diplomacy only by self-play, without fine-tuning or regularizing on human data. The method simplifies the previous architecture from the PIKL/CICERO papers by FAIR (https://openreview.net/forum?id=F61FwJTZhb) by avoiding using reinforcement learning and human demonstrators.

Given that the authors use commercially available LLMs (GPT4 and LLAMA) I contribute the success of the method to the alignment of the LLMs to human preferences and communication style. Additionally, they exploit the planning capabilities of current LLMs to bypass the need for RL.

**Strengths:**

1. The introduction of sub-goals is an interesting feature. It acts as in a chain-of-thought mechanism, which is proven to improve the LLM output and avoid hallucinations. Smaller reasoning steps are more robust than long-horizon planning.

2. The paper proposes a novel architecture with a memory buffer that acts like a pseudo-RAG to improve the context of the LLMs information.

3. The authors provide a very complete repo to reproduce the experiments.

**Weaknesses:**

1. Well, the paper benefits from the improvement of LLMs over time, as compared with GPT2 (used by CICERO). The alignment to human data is not needed as now the LLMs have the alignment incorporated by the vendor. Same for the planning step, the LLMs intrinsically got better at planning so now we can demote RL for free.

2. When evaluating against CICERO, the win rate over CICERO is not that high with your model winning the game only (<1%) of the time. I believe the win rates are not impressive, but the fact that the victories are achieved by a more lightweight model that is cheaper than training CICERO. As it doesn't require human data.

3. The alignment with humans playing and conversational style is hard to assess as it will require having your model to play against humans in an online platform and assessing whether players realize they were playing against an AI player.

**Questions:**

1. See weakness above.

**Limitations:**

The authors have not provide any analysis on the model limitations.

---

> ### Author Rebuttal · Authors · 2024-08-07
>
> > Well, the paper benefits from the improvement of LLMs over time, as compared with GPT2 (used by CICERO). The alignment to human data is not needed as now the LLMs have the alignment incorporated by the vendor. Same for the planning step, the LLMs intrinsically got better at planning so now we can demote RL for free.
>
> **A:** We appreciate your point regarding the improvements in LLMs and their impact on our work. As illustrated in Figure 5, while the enhanced alignment in LLMs indeed boosts performance (GPT-4 is better than others), we observed that a vanilla GPT-4 still falls short in AI diplomacy without our framework (Table 2). This indicates that the alignment in LLMs lays a foundation, but our approach is key to unlocking the models' potential in social simulation.
>
>
> > When evaluating against CICERO, the win rate over CICERO is not that high with your model winning the game only (<1%) of the time. I believe the win rates are not impressive, but the fact that the victories are achieved by a more lightweight model that is cheaper than training CICERO. As it doesn't require human data.
>
> **A:** We agree that the main contribution of this work is providing a low-cost paradigm to address AI diplomacy without human data. In the revision, We will highlight the data efficiency in the experiment section. Moreover, we observed a large gap by comparing the scores the agents gained. Our agent's score is about 10% higher than Cicero's.
>
> > The alignment with humans playing and conversational style is hard to assess as it will require having your model to play against humans in an online platform and assessing whether players realize they were playing against an AI player.
>
> **A:** Thanks for your suggestion. Considering the ethical issue, we will conduct experiments to play with human players in our future work after getting official permission. You can refer to the dialogue shown in Fig. 6~7 and the one-page PDF for more vivid conversation examples.

---

> > ### Comment · Reviewer_36qq · 2024-08-07
> > **Improved rating:7**
> >
> > I would like to thank the authors for clarifying my questions and engaging with interest. Since my concerns were clarified I have upgraded my rating to 7. This is a good paper that merits an accept due to its contributions over a widely researched topic such as Diplomacy and negotiation.

---

> > > ### Author Response · Authors · 2024-08-08
> > > **Appreciation for Your Support**
> > >
> > > Thank you for your kind words. We sincerely appreciate your positive feedback and the improved rating. Your acknowledgment of our work serves as a strong motivation for us to continue striving for excellence in our future work.

---

> > > ### Author Response · Authors · 2024-08-09
> > > **Request for Verifying Rating Update**
> > >
> > > We are writing to express our sincere gratitude for your positive feedback and upgraded rating to **7**.
> > >
> > > However, we have noticed that **the system remain reflects the previous rating of 6**. We understand that such oversights can happen, and we are reaching out to kindly bring this to your attention.
> > >
> > > Could you please verify if there has been any update or if there is a need for further action on your part to ensure that the revised rating is accurately reflected in the system? Your assistance in this matter is greatly appreciated.
> > >
> > > Thank you once again for your valuable feedback and for your understanding.

---

### Official Review · Reviewer_raEc · 2024-07-05

**Soundness:** 2
**Presentation:** 2
**Contribution:** 2
**Rating:** 4
**Confidence:** 4

**Summary:**

This paper focuses on diplomatic activities using LLM-based societal agents without relying on human data. It introduces a new paradigm for AI diplomacy agents that can improve through self-play and experience collection. The new agent, Richelieu, achieves state-of-the-art performance compared to current methods and demonstrates generalizability across different LLMs.

**Strengths:**

This paper presents a new AI agent for diplomatic scenarios that surpasses all previous methods. This agent can improve autonomously without relying on human data. And the accompanying figures effectively illustrate the concept.

**Weaknesses:**

- There are some typos (line 61, line 69), and certain citations (line 80, line 237, line 276) are not properly functioning.
- Some experimental details are still unclear.
    - Which base model is utilized in the experiment for Figure 4 and Table 1? Is the improvement over all other baselines attributed to the agent paradigm or to the strong capabilities of GPT-4 or LLaMA3 compared to previous base models like Cicero, which only used a 2.7B model?
    - In the main experiment for Table 1, when Richelieu plays as a randomly selected country, what model or agent represents the other countries?
    - In the ablation study, what are the effects of blocking each module, such as only blocking the sub-goals module?
- Techniques such as memory and planning in building agents have been proposed previously, which reduces the novelty of the method.

**Questions:**

- Is the social reasoning flow the same or different for the planner and negotiator parts?
- Can this paradigm generalize to other fields in social simulation?

---

> ### Author Rebuttal · Authors · 2024-08-07
>
> > There are some typos (line 61, line 69), and certain citations (line 80, line 237, line 276) are not properly functioning.
>
> **A:** We will fix them in the revision.
>
> > Which base model is utilized in the experiment for Figure 4 and Table 1?
>
> **A:** We use GPT 4 as the base model for our agent in the experiment.
>
> > Is the improvement over all other baselines attributed to the agent paradigm or to the strong capabilities of GPT-4 or LLaMA3 compared to previous base models like Cicero, which only used a 2.7B model?
>
> **A:** Despite the improvement in the capabilities of the large language model (LLM) being of significant help to the enhancement of our model's capabilities, merely relying on the LLM cannot achieve very good results. When we initially used the LLM directly, it was unable to make correct decisions. Our ablation study (line 315) showed that as our model improved, its negotiation and reasoning abilities also improved (Table 2).
>
> At the same time, we also analyze the generalization of our framework by using different LLMs. The results (Figure 5) showed that despite the differences in the final outcomes due to the varying capabilities of the different LLMs, the self-evolving mechanism can steadily improve the agents' performance.
>
> Cicero used extra human data for training the dialogue and planning modules separately. In contrast, Richelieu relies on no human data, and a direct application of LLM e.g. GPT-4 is unsuccessful as the experiment shows. It is the proposed combined paradigm that enhances LLM models like GPT-4 to achieve complex planning capability, as demonstrated by the experiments.
>
> > In the main experiment for Table 1, when Richelieu plays as a randomly selected country, what model or agent represents the other countries?
>
> **A:** In the experiment, we randomly select 3 or 4 countries to be played by Richelieu, and the other countries will be controlled by Cicero.
>
> > In the ablation study, what are the effects of blocking each module, such as only blocking the sub-goals module?
>
> **A:** Without a sub-goal, Richelieu will be very shortsighted and will tend to favor short-term gains over long-term gains. In the planner, to formulate a sub-goal, Richelieu needs to consider long-term benefits. Therefore, making decisions based on the sub-goal can naturally ensure that the decision includes consideration of long-term benefits. But after blocking this module, most of the decisions made by Richelieu are to occupy as much territory as possible in the current turn, similar to a greedy algorithm. Moreover, Richelieu's ability to handle relations with other countries will decline.
>
> > Techniques such as memory and planning in building agents have been proposed previously, which reduces the novelty of the method.
>
> **A:** The memory and planning modules do not work well without the self-play data and reflection mechanism for LLM agent models as shown in Table 2. Such an integrated self-evolving scheme for LLM agents achieves high performance without human data, which has not been verified in previous work on AI diplomacy.
>
> > Is the social reasoning flow the same or different for the planner and negotiator parts?
>
> **A:** A major difference between our model and traditional models is that there is no need to separately establish a decision-making model and a negotiation model. Therefore, in our model, the social reasoning result applied in the process of planning and negotiation is the same. Our model will perform social reasoning at the beginning of each turn. We will analyze the current state, the strengths and weaknesses of each country, and infer the strategic intentions of each country. Based on these, we will speculate which country can be our potential ally and which country will be our adversary. The social reasoning result will first be used to establish its own sub-goal. Then, during the negotiation phase, negotiations will be conducted based on the results and goals. During the negotiation process, the other party's words may cause us to modify the results.
>
>
> > Can this paradigm generalize to other fields in social simulation?
>
> **A:**  Our framework is capable of applying to most social interaction tasks. Most components in our framework can be easily generalized to a new task by modifying the content. Social reasoning enables the agent to handle complex and dynamic social relationships. The negotiation pipeline opens the potential of communicating with others to prob the other's mind or reach a consensus. The hierarchical strategy with reflection enhances the ability to handle long-term planning. The self-evolving mechanism (reflection with self-play memory) further improves the overall performance without manual supervision. These modules cover most of the challenges in multi-agent interactions, e.g., werewolf games, economic games, and daily interactions.
>
> We further adopt our framework to a ***werewolf game***. The results demonstrate our reasoning framework achieves comparable results to the other methods. Due to the time limitation, we do not apply the self-play game in the current version. To be specific, in the experiment, we let our agent play as a werewolf in a seven-player game, where there are two werewolves, one witch, one seer, one guard, and two villagers. The experimental results show that the win rate of our agent is 59.2%. For comparison, the specifically designed LLM-based agent achieved ~65% win rate.

---

> ### Author Response · Authors · 2024-08-14
> **Looking forward to your feedback**
>
> Dear Reviewer raEc and EuFx,
>
>     I hope this message finds you well. As the author discussion period is nearing its end, we kindly request the opportunity to address any further feedback you may have regarding our response to submission 3239.  We have made substantial clarifications and provided additional results based on your initial comments. If you have any further questions or require clarification on our response, please let us know. Your insights are crucial for the improvement and assessment of our work. Considering the final score, we kindly ask for a potential score improvement if you believe our response has addressed your major concern.
>
>     We greatly appreciate your time and expertise and look forward to your response.
>
> Best regards,
>
> Authors of Submission 3239

---

### Official Review · Reviewer_HUgn · 2024-07-13

**Soundness:** 3
**Presentation:** 3
**Contribution:** 3
**Rating:** 6
**Confidence:** 3

**Summary:**

The paper presents “Richelieu,” a self-evolving large language model (LLM)-based agent designed for the game of Diplomacy. Richelieu integrates strategic planning, social reasoning, and memory reflection to handle complex multi-agent environments without relying on domain-specific human data. The model self-evolves through self-play games, demonstrating its effectiveness.

**Strengths:**

1. This paper studies an interesting social science problem and introduces an LLM-based paradigm to build an AI Diplomacy agent.
2. This paper proposes a self-evolve strategy through self-play without human data.
3. The presentation of this paper is good, clearly describing the core idea of the paper.

**Weaknesses:**

1. Lack of in-depth analysis of the middle process. In the proposed framework, the Social Reasoning and Planner with Reflection play an important role. For example, I wonder if the LLM can accurately model relationship and inferring intention. The author could provide more analysis or examples of these modules to prove the effectiveness of LLM on this task.
2. Some technical details are not clear. For example, in the process of experience retrieval, both object similarity and state similarity are considered, but no descriptions of their implementations.
3. While the paper demonstrates the effectiveness of Richelieu, scalability to larger and more diverse environments remains to be fully explored.

**Questions:**

1. The generation process of LLM is uncertain, which may lead to unstable reasoning and reflection, and finally inconsistent results. Does this paper consider this problem and how to deal with it?
2. Could you further elaborate on the significance of self-play games in the self-evolution process?
3. How to consider both object similarity and state similarity in the process of experience retrieval?
4. Could you provide examples of how Richelieu adapts its social reasoning and negotiation tactics based on past interactions?

**Limitations:**

See Weaknesses and Questions

---

> ### Author Rebuttal · Authors · 2024-08-07
>
> > Lack of in-depth analysis of the middle process. If the LLM can accurately model relationship and inferring intention. The author could provide more analysis or examples of these modules to prove the effectiveness of LLM on this task.
>
> **A**: The LLM can accurately model relationships and inferring intention. We conduct an experiment to evaluate the success rate that the agent can successfully identify the social relationship and infer others' intentions. As the baselines do not explicitly model the relationship and intention, we can not directly access the ground truth for evaluation. Instead, we let all players use our agent but with different LLMs, i.e., 4 countries use GPT-4 and 3 countries use Llama3. The accuracy is reported in the following:
>
> |     | GPT-4  |Llama3|
> |  ----  | ----  |----|
> | relationship  | 85.74% |85.52%|
> | intention (sub-goal)  | 74.67% |74.11%|
>
> We can see that the accuracy of social reasoning is consistent with the overall performance of the agent, indicating the effectiveness of social reasoning. In the one-page PDF, we further provide an example case to demonstrate the effect of the self-play games. It shows that the agents evolved by reflection with memory collected in the self-play games can handle a long-term planner.
>
>
> > How to consider both object similarity and state similarity in the process of experience retrieval?
>
> **A**: The overall similarity is the weighted sum of the two similarity metrics.
> $$ S = \lambda S(s_t,s_{t'})+(1-\lambda)S(\chi_{i,t},\chi_{i,t'}) $$
> In our implementation, $\lambda=0.65$. $s$ is the description of state and $\chi$ is the sub-goal at corresponding state. we select the top $m$ experiences from the memory buffer with the highest similarity $S$ to the current turn. Historical turns similar to the current turn state and sub-goal can reflect and optimize the decisions made in the current turn. We will add more details in the revision.
>
> > Scalability to larger and more diverse environments remains to be fully explored.
>
> **A**: Thanks for your suggestion. Our framework is capable of applying to most social interaction tasks. Most components in our framework can be easily generalized to a new task by modifying the content. Social reasoning enables the agent to handle complex and dynamic social relationships. The negotiation pipeline opens the potential of communicating with others to prob the other's mind or reach a consensus. The hierarchical strategy with reflection enhances the ability to handle long-term planning. The self-evolving mechanism (reflection with self-play memory) further improves the overall performance without manual supervision. These modules cover most of the challenges in multi-agent interactions, e.g., werewolf games, economic games, and daily interactions.
>
> We further adopt our framework to a ***werewolf game***. The results demonstrate our reasoning framework achieves comparable results to the other methods. Due to the time limitation, we do not apply the self-play game in the current version. To be specific, in the experiment, we let our agent play as a werewolf in a seven-player game, where there are two werewolves, one witch, one seer, one guard, and two villagers. The experimental results show that the win rate of our agent is 59.2%. For comparison, the specifically designed LLM-based agent achieved about 65% win rate.
>
> > The generation process of LLM is uncertain, which may lead to unstable reasoning and reflection, and finally inconsistent results. Does this paper consider this problem and how to deal with it?
>
> **A**: In experiments, we set a temperature of 0.3 to ensure a relatively stable generation of LLM policies. The overall reasoning framework also ensure the stability and consistency in the AI agent's performance. Besides, we find that the state-of-the-art LLM (GPT-4 or Llama 3) can deal with this problem well, as is shown in Fig. 5.
>
>
> > The significance of self-play games in the self-evolution process.
>
> **A**:
> The reflection module highly relies on historical experiences to guide the generation of effective sub-goals. Thus, the diversity of the memory will lead to the success of the reflection. The self-play games can help the agent autonomously explore different experiences and collect them in the memory, which is fundamental for the whole self-evolving process. In this way, we can build an agent without human training data or any existing agents for the task. The results in Figure 5 show that as self-play progressed, Richelieu's win rate continued to increase until it reached a stable value. Its effectiveness can also be verified by the results of the ablation study (Table 2). Moreover, we also provide an example in the one-page PDF, showing that self-play memory can guide the agent to consider the long-term effect of the strategy.
>
> > **Could you provide examples of how Richelieu adapts its social reasoning and negotiation tactics based on past interactions?**
>
> **A**: Examples are given in the one-page PDF.

---

> > ### Comment · Reviewer_HUgn · 2024-08-11
> > **Reply**
> >
> > Thank the authors for your clarifications. I will increase my score from 5 to 6.

---

> > > ### Author Response · Authors · 2024-08-13
> > > **Thanks**
> > >
> > > We sincerely appreciate the insights you’ve shared for this work and are grateful for your consideration in raising the score.

---

### Official Review · Reviewer_EuFx · 2024-07-13

**Soundness:** 2
**Presentation:** 2
**Contribution:** 2
**Rating:** 4
**Confidence:** 4

**Summary:**

This paper propose a new framework for LLM-based agents to play diplomacy games and improve themselves. The proposed framework, Richelieu, have several components and many abilities. The authors perform good experiments and ablation study to show how the framework.

**Strengths:**

- The framework have several core components, and I believe the authors build a good agent that can handle the game.
- The game or the setting is good for evaluating the comprehensive ability of LLM-based agents.

**Weaknesses:**

- What does "evolve" mean in the article? Does it only refer to the storage of memory modules? If so, it seems that the model itself has not been updated.

- The baseline in the article is not a LLM-based agent, can a stronger baseline be provided? For example, ReAct, PlanAct, AutoGPT.

- The article lacks some references to work in the field and discussion of related work, e.g., [1][2][3].

[1] The Rise and Potential of Large Language Model Based Agents: A Survey

[2] ReAct: Synergizing Reasoning and Acting in Language Models

[3] ProAgent: Building Proactive Cooperative AI with Large Language Models

**Questions:**

See Weaknesses

**Limitations:**

- The self-evolve in the paper is not clear.
- For the memory module and self-play module, a more detailed analysis should be included.

---

> ### Author Rebuttal · Authors · 2024-08-07
>
> > What does "evolve" mean in the article? Does it only refer to the storage of memory modules? If so, it seems that the model itself has not been updated.
>
> **A**: "evolve" means the AI agent's capability and strategy are autonomously enhanced over time without direct human supervision. Beyond memory storage, self-evolution is achieved through several key components, including the self-play game to generate diverse memory, the experience abstract to extract meaningful information from memory, and the reflection mechanism to update the planner. Besides, the introduced thought flow also plays an important role in the overall performance. Thus, we do not need update the parameters of neural network of the base LLMs. We argue such a parameter-free evolution mechanism is more efficient than finetuning the LLM.
>
> > The baseline in the article is not a LLM-based agent, can a stronger baseline be provided? For example, ReAct, PlanAct, AutoGPT.
>
> **A**: Thanks for your suggestion. As there were no previous works that explored the use of LLM-based agents for AI diplomacy, most of the baselines are RL-based agents. The ablation methods in Table 2 can be regarded as LLM-based baselines by combining different techniques, such as chain of thought, reflection, memory, and etc. We can see that deploying vanilla LLM or common techniques to build an LLM-based agent can not work well in the task. Following your suggestion, we further built an LLM-based agent using AutoGPT and compared it with our agent. In the testing, we randomly select three countries to be controlled by Richelieu, and the other four countries to be controlled by AutoGPT. Note that the agent controls each country independently. The results are given below.
> |     | win  | most SC | survived|defeated|
> |  ----  | ----  | ----  | ----  | ----  |
> | Richelieu_1  | 9.3%|18.2%|37.9%|34.6%|
> | Richelieu_2  | 9.9%|19.4%|37.7%|33.0%|
> | Richelieu_3  | 8.1%|17.4%|39.2%|35.3%|
> | AutoGPT_1  | 1.2% |4.6%|32.4%|61.8%|
> | AutoGPT_2  | 1.2% |4.2%|34.4%|60.2%|
> | AutoGPT_3  | 1.5% |4.0%|32.5%|62.0%|
> | AutoGPT_4  | 2.6% |3.6%|32.3%|61.5%|
>
> |     | win  | most SC | survived|defeated|
> |  ----  | ----  | ----  | ----  | ----  |
> | Richelieu  | 9.10% |18.33%|38.27%|34.30%|
> | AutoGPT  | 1.63% |4.10%|32.90%|61.38%|
>
>
> > The article lacks some references to work in the field and discussion of related work, e.g., [1][2][3].
>
> **A**: We will add more references in the revision. Most previous work focuses on developing the reasoning framework for LLM-based agents, such as ReAct[2], PlanAct, and AutoGPT, to accomplish complex **single-agent** tasks. ProAgent[3] provides a framework for decentralized multi-agent collaboration. In this setting, it assumed the relationship between agents is fixed (collaborative) and the agents do not need to negotiate with others. Differently, AI diplomacy is more challenging, where the relationship is uncertain and dynamically changing, and the agents need to actively negotiate with others and plan long-term strategies to win the game. Hence, the previous work can not directly apply to the task, and we further introduce a new self-evolving LLM-based agent for AI diplomacy. Note that the proposed approach is not limited to specific LLMs or tasks, but a principled framework to enable LLM-based agents to work in complex environments with social interactions.
>
> > For the memory module and self-play module, a more detailed analysis should be included.
>
> **A**: The memory module will record the state of each turn, as well as the negotiation results and final actions taken by each country. It will also record the state changes over a period of time after this turn, thus determining the impact of the actions. Therefore, we can use the memory to find turns similar to the current turn and reflect and optimize based on the actions taken and subsequent state changes.
> As self-play progresses, the experiences in the memory modules will accumulate. We can find more similar historical turns as experiences during reflection, thereby enhancing the capabilities of our model.
>
> Based on the results from Table 2 and Figure 5, we can see that the memory module and self-play have a significant impact on enhancing the model's capabilities. Moreover, as self-play progresses, the model's capabilities gradually improve and eventually reach a relatively stable level.

---

> > ### Comment · Area_Chair_LnNV · 2024-08-12
> > **Reviewers EuFx and raEc**
> >
> > The authors have responded to your reviews. Have they sufficiently answered your questions, and if not, do you have any further clarifying questions you would like to ask? The author discussion period ends tomorrow (13th).

---

### Author Rebuttal · Authors · 2024-08-07

Our rebuttal includes a one-page PDF and the following four rebuttals for each Official Review. The PDF contains an example. We show two cases with similar states: the first shows decisions and negotiations made without self-play, and the second shows those made after self-play. This example shows that self-play influences the model's decision-making and negotiation, making it focus more on long-term benefits.

---

### Decision · Program_Chairs · 2024-09-25

**Decision:**

Accept (poster)

**Comment:**

This paper describes an LLM agent that can solve the game of AI Diplomacy only by self-play, without fine-tuning or regularizing on human data. The paper has several strengths. First, it studies an interesting social science problem. Second, it proposes a novel LMM-based agent architecture that uses self-learning together with a memory that acts as a type of RAG to improve its performance at solving the problem (playing the game) over time. The method simplifies the previous agent architecture for solving this problem from the PIKL/CICERO papers, providing a lower-cost solution that does not require human data and outperforms the prior approach. Finally, the paper is well written and the presentation is mostly easy to follow.

A number of weaknesses of the paper were also identified. First, the fact that newer LLM models are used in the empirical analysis raises the question of how important is this to the outcomes obtained, and the relative impact of this versus the proposed new memory and planning techniques added is not adequately explained. The authors' rebuttal provided some discussion and evidence that even newer LLM models cannot solve the problem alone, and should this paper be accepted, the paper should be revised to emphasize that the choice of LLM used does not impact the significance of the work. Second, techniques for using memory and planning have been used before, which lessens the novelty of the approach. Finally,  the self-evolve aspect of the approach is not sufficiently explained. This last point was also adequately addressed in the authors' rebuttal, and should be incorporated in the final version if the paper is accepted.